# A Comparison of the Influence of Vegetation Cover on the Precision of an UAV 3D Model and Ground Measurement Data for Archaeological Investigations: A Case Study of the Lepelionys Mound, Middle Lithuania

**DOI:** 10.3390/s19235303

**Published:** 2019-12-02

**Authors:** Algimantas Česnulevičius, Artūras Bautrėnas, Linas Bevainis, Donatas Ovodas

**Affiliations:** Department of Cartography and Geoinformatics, Institute of Geosciences, Faculty of Chemistry and Geosciences, Vilnius University, LT-03101 Vilnius, Lithuania; arturas.bautrenas@gf.vu.lt (A.B.); linas.bevainis@gf.vu.lt (L.B.); ovodas@gmail.com (D.O.)

**Keywords:** GPS measurement, UAV, 3D models, measurement precision

## Abstract

The aim of this research was to conduct a comparative analysis of the precision of ground geodetic data versus the three-dimensional (3D) measurements from unmanned aerial vehicles (UAV), while establishing the impact of herbaceous vegetation on the UAV 3D model. Low (up to 0.5 m high) herbaceous vegetation can impede the establishment of the anthropogenic roughness of the surface. The identification of minor surface alterations, which enables the determination of their anthropogenic origin, is of utmost importance in archaeological investigations. Vegetation cover is regarded as one of the factors influencing the identification of such minor forms of relief. The research was conducted on the Lepelionys Mound (Prienai District Municipality, Lithuania). Ground measurements were obtained using Trimble GPS, and UAV “Inspire 1” was used for taking aerial photographs. Following the data from the ground measurements and aerial photographs, large scale surface maps were drawn and the errors in the measurement of the position of the isolines were compared. The results showed that the largest errors in the positional measurements of fixed objects were conditioned by the height of grass. Grass with a height of up to 0.1 m resulted in discrepancies of up to 0.5 m, whereas grass that was up to 0.5 m high led to discrepancies up to 1.3 m high.

## 1. Introduction 

During the initial stage of an archaeological investigation, one of the most important principles is to identify a potential object, to determine its boundaries and area. Traditionally, large-scale topographic maps and geodetic measurements are widely used during the initial stage of reconstruction. 

Aerial photographs were mainly used where archaeological sites coincided with the areas covered by aerial topography and only on fragmentary basis due to their high cost. High-resolution space images have only become possible within the last decade, but they do not cover continuous areas. Moreover, high resolution photos are not always available for academic research or studies. At the beginning of the 21st century unmanned aerial vehicles, better known as drones, were employed to identify and map potential archaeological objects. They have a number of advantages which include the following characteristics: low price, high resolution, large scale, and multispectral. A very important advantage of unmanned aerial vehicles is the creation of 3D models using photogrammetric techniques. These 3D models reveal the small roughness of the surface. Such alterations in the surface serve as identifiers, when searching for potential archaeological sites. 

The issues of reliability and accuracy of aerial photographs obtained using unmanned aerial vehicles have already been addressed in studies by many academic researchers [1,2,3,4,5,6,7,8,9,10,11,12,13,14,15,16,17,18,19,20,21]. The use of unmanned aerial vehicles provide a fast and inexpensive way to explore ground surface and to identify objects of interest [22], however, research on assessing the precision of aerial images from unmanned aerial vehicles is scarce [23,24,25,26]. The accuracy of aerial images produced with the help of unmanned aerial vehicles (UAVs) can be affected by a number of factors, for example, altitude of flight, the image quality of the photo camera, the design of the UAV route, the methods of georeferencing, and others. An appropriate design of the UAV route ensures cruise altitude and constant aerial image coverage of the whole territory. An appropriate project for the flight and a high-quality photo camera effect the efficiency of photogrammetric processing of the images obtained. Further investigations are simplified by using the well-tested and broadly applied mathematical and photogrammetric algorithms for image processing. The problems occur while designing a 3D model of the territory captured in aerial images. The initial 3D model is created in the conditional coordinate system, which is later linked to the officially used coordinate system. The coordinates can be connected in one of the following two ways: by direct graphical connection of the position of the object in the aerial image to the coordinate system (less precise) or by linking the GPS measurements of fixed objects to the coordinates of the aerial images. The accuracy of the vertical positioning of given points is a highly important factor in designing the 3D relief models that are used for identification, analysis, and mapping of archaeological objects. 

Recent research [18,19,20,21,22,23,24,25,26,27,28,29,30,31,32] has shown that while aiming for high accuracy of the vertical positioning of the objects, it is not enough to use a global navigation satellite system (GNSS); ground control points (GCPs) have to be applied as well. Such a combined technique allows for the design of a more accurate digital relief model (DRM), where the precision of vertical positioning of points equals 0.7 cm. 

The aim of this research is to conduct a comparative analysis of the precision of ground geodetic measurements and aerial photographs from an unmanned aerial vehicle, while establishing the positional accuracy of the identified objects. The archaeological objects of the Middle Ages in the eastern coast of the Baltic sea are often related to natural relief forms, which were modified by people while building fortifications and settlements around them [33,34,35,36,37]. These archaeological objects are now in forests, agricultural lands, and urbanized territories. The surface of archaeological objects in such urban territories has been exposed to significant changes or has been fully destroyed. The use of aerial images from unmanned aerial vehicles for the positional identification of archaeological objects is highly limited. Due to dense vegetation and the foliage of tall trees, the application of aerial imaging in wooded territories is restricted. The surface of archaeological objects in agricultural territories is partially extant. Therefore, aerial images can be rather efficient in seeking to identify positions of archaeological objects in meadows and woodless territories. 

The narrow spectral and surface thermal analysis methods are applied for the investigation of the structural diversity of vegetation cover on the basis of UAV aerial images [38,39,40,41,42]. Studies have mainly focused on the influence of big ligneous plants on the mapping of surface elements, whereas the impact of low herbaceous vegetation on low forms of archaeological relief has, so far, not been exhaustively researched [43]. Our research aims to assess the quality of aerial images, ultimately seeking to design accurate digital 3D relief models for the identification of archaeological objects [44,45,46,47,48,49,50].

For the identification of small surface irregularities (small archaeological objects) we applied the computer program “Circle_3p”, developed by the Department of Cartography and Geoinformatics, Vilnius University, applying the classical Delaunay method (author Artūras Bautrėnas). The results of the study showed that this method is effective in grassy mounds. 

## 2. Research Object, Materials, and Methods

The object of the research is the Lepelionys Mound, which is located in the Prienai Administrative Region of Kaunas County (Figure 1). It dates back to the second half of the first millennium. At the beginning of the second millennium a settlement was established there, covering an area of 9 hectares around the mound. The Lepelionys Mound is on the left side of the road from Vilnius to Prienai (60 km to the west of Vilnius). The territory of the ancient settlement is on both sides of the road, but its bigger part is located on the left side. The Vilnius-Prienai road was built in the second half of the 20th century. While designing the road, the relief of the former ancient settlement was affected but some small and low relief forms of anthropogenic origin still remain, dating back to between the 9th and 12th centuries [51,52]. The main archaeological object, the Lepelionys Mound, was investigated by archaeologists in the second half of the 20th century. During these archaeological investigations the territory boundaries and the protection zone of the ancient settlement were distinguished (Figure 1). 

Ground geodetic measurements and photos taken by the camera on the unmanned aerial vehicles were applied while designing the three-dimensional relief models. Comparisons of accuracy between the UAV 3D model and the ground measurements of the Lepelionys mound were carried out twice, in August 2018 and June 2019.

Ground geodetic measurements were carried out with a Trimble R4 GPS device (measurable accuracy in favorable conditions: X, Y is set to ±8 mm and Z to ±15 mm) on 9 August 2018. Since the mound is in a fully open area and not covered by buildings or greenery (Figure 1), the measurements were collected with maximum accuracy. During the collection of these measurements, the coordinates of 212 characteristic ground-surface points were recorded. After analyzing the accuracy of the measured point coordinates, 179 points were mapped to the LKS-94 coordinate system (Figure 2). 

Since the topographic photograph can be used to estimate the accuracy of the aerial photographs, 10 ground control points (GCPs) were measured in parallel to the ground points (Figure 3). 

Figure 4 shows two objects, the coordinates of which were used for creating the aerial photograph model.

The vegetation is one of the most important indicators of archaeological objects. Information on human activities is reflected in the variation of the lushness of vegetation. Homogeneous vegetation is characteristic of the investigated territory, since for several decades most of the surroundings of the mound have been used as pasture. Local differences in herbaceous vegetation in the mound surroundings over a long period of time have been predetermined by changes in the surface relief layer caused by the following human activities:
(i)Organic waste was thrown at the foot of the mound;(ii)In the territory of the ancient settlement the ground was excavated for substructures of buildings and the soil (sediment) was poured beside the walls of the building;(iii)The ancient settlement was surrounded by palisades, the stakes of which were driven into the ground and the excavated soil fortified the foundation of the fence;(iv)Organic and mineral waste (ceramic fragments, bones of the animals used for food, worn out shoes, and clothes) was thrown over the palisade of the settlement.

All the aforesaid factors resulted in physical differences in the present vegetation, i.e., lusher or sparser vegetation. It is important to point out that currently there is a pasture in the former territory of the settlement, where grazing starts at the end of April and lasts until October. The whole area is grazed in this time and the anthropogenic impact on the surface was equal during photofixation in August. 

The picture in Figure 5 provides a visual representation of the camera sensor and the field of view. Using the width of the camera sensor, the focal length, and the drone altitude the ground sample distance (GSD) can be calculated (Figure 5).

The equation we use to calculate the *GSD* is:
(1)$$GSD\;=\;\frac{\left(sensors\;width\;\times \;altitude\;\times \;100\right)}{\left(focal\;length\;\times \;image\;width\right)}$$

Photofixation of aerial images was conducted using the unmanned aerial vehicle (UAV) INSPIRE 1. Its technical parameters are presented in Table 1.

The front overlap of the pictures taken is 80% and the side overlap is 70%. The “double grid” mission flight plan was used for a more detailed and accurate 3D model. The flight was made at the height of 50 m, therefore, respectively, the GSD equals 1.7 cm. 

There were 199 photos that were processed with special photogrammetric “Pixoprocessing” software. The point cloud, the digital surface model (DSM), and the orthomosaic were obtained during this process.

The study included an assessment of the mismatches between the elevation isoline positions acquired from the ground geodetic measurements and from the aerial images from the UAVs. An associate professor of the Department of Cartography and Geoinformatics, Artūras Bautrėnas, designed the computer program “Circle_3p”, which employs the classical method of Delaunay and ensures a consistent systemic selection of points. Using the Delaunay triangulation method, altitude interpolation of the ground measurement points was performed and an isoline view was generated. An analogous method was used for the interpolation of the elevation of surface points and the generation of isolines using the images taken by the camera on the UAV (Figure 6). The following indicators were calculated: ±N_i_ which is the sequence number of the analyzed point in the positive or the negative deviation from the base (ground geodetic measurement) isoline, ±ΔS_i_ which is the length of the perpendicular to the positive or the negative side of the analyzed point, ±ΔZ_i_ which is the calculated correction of the overdose to the positive or the negative side, and ±D which is the distance between the base (ground geodetic measurement) isoline point and the UAV isoline point.

For the calculation of the deviation of the target position the following formula was used:
(2)$$\pm \Delta S_{i}\;=\;\left(y_{\mathrm{Ni}}\;-{\;y}_{\mathrm{Ti}}\right)\cos\alpha \;-\;\left(x_{\mathrm{Ni}\;}-{\;x}_{\mathrm{Ti}}\right)\sin\alpha .$$
where α is the directional angle of the segment N_i_–N_i + 1_, T_i_ is the number of the interpolated UAV measurement point, and i is the number of the point for each fragment of the ground geodetic and UAV isolines. 

As we know what the isoline step is (0.5 m), the distance between the horizontal (*±D*) at each N_i_ point can be calculated by geometric interpolation. The difference in height ± Δ*Z_i_* is calculated using the formula:
$$\pm \Delta Z_{i}\;=\;\frac{\pm h}{\pm D_{i}}\;\times \;\pm \Delta S_{i}$$
where *h* is the isoline step, *D* is the distance between the horizontal at the calculated point, and the ± sign depends on the direction of the horizontal deviation.

## 3. Results

### 3.1. Creating a Two-Dimensional (2D) Relief Model 

In order to confidently state that the 2D relief model is sufficiently precise and can be used as a benchmark for estimating the models, which were made by using aerial photometric methods, the horizontals were drawn automatically in accordance with strict interpolation rules. 

In order to perform the automated relief modelling, it was necessary to select pairs of measured points, among which it would be possible to calculate the exact horizontal surfaces of the relief, i.e., interpolate heights. Therefore, the Delaunay triangulation method was chosen to interpolate the heights [53]. 

### 3.2. Drawing of a Topographic Plan 

First, the Delaunay triangulation (Figure 7) was completed among 179 selected topographic points using the program “Circle_3p”. This allowed 321 triangles to be selected, among which the interpolation of the triangle vertices was performed. 

Among these triangle vertices, horizontal interpolation was performed in the LAS07 height system using a selected step of 0.5 m (Figure 8). The coordinates of 1676 extra points, plotted as horizontals, were calculated during the interpolation. 

The interpolation points were uploaded to TopoPlan (AutoCAD 2016). The horizontals were plotted using the “Spline” function (Figure 9 and Figure 10). 

The cross-sections of the Lepelionys Mound were created with the help of aerial images taken by the camera on UAV, which highlighted the minor anthropogenic forms of relief on the slope of the mound, i.e., the remains of the former tree trunk wall (Figure 11).

## 4. Discussion

### 4.1. Evaluation of the Precision of theAaerial Images 

One of the most time-consuming tasks in aerial photography is to set out the GCPs and to coordinate them. Therefore, it is necessary to find the optimal number of GCPs in order to minimize the preparatory work. It should also be possible to estimate the feasible use of coordinated stable land objects (Figure 4) instead of bearing marks, which would further simplify the preparatory work. Therefore, ten ground control points (marks) in the study area are used to estimate the accuracy of the coordinates of 10 objects in the study area (Figure 3). 

In order to evaluate the accuracy of the 3D model, it was created incorporating all ten marks and the coordinates of all the objects were measured in this model. The differences between the coordinates of objects in the 3D model and the coordinates measured from the topographic image do not exceed the double (Trimble GPS) accuracy (Table 2 and Table 3) for those objects that are clearly seen in the 3D model (np-508, -515, -582, and -587). The accuracy of the other objects is poorer due to the vegetation (grass), which complicates their identification. 

The random error distribution depends on the accuracy of the object identification, and therefore the graph consists of taking the errors in absolute size in mm.

The error analysis shows that they increase significantly when the orientation marks are fewer than five (Table 5, Figure 5), even for those objects that are visible in the 3D model (np-508, -515, -582, and -587). Therefore, it can be argued that in order to maintain the accuracy of measurements, there should be at least five orientation marks. It has been noticed that the error rate is influenced not only by the vegetation but also by the experience and thoroughness of the operator measuring the 3D model. As the precision of the well-known objects is practically unchanged (from ten to five marks), it can be argued that the 3D model should operate with maximum accuracy with five GCPs and the use of easily visible coordinated objects (Figure 12, Table 2, Table 3, Table 4 and Table 5). 

As seen in Table 3, the random error distribution depends on the accuracy of the object identification. Similarly, the absolute errors of objects have been calculated for the 3D models with different number of ground control points (Table 4 and Table 5). 

### 4.2. Evaluation of the Influence of Vegetation Covers

In 2019, a comparison of the Lepelionys mound surface isolines obtained using the UAV 3D model or the ground measurements showed that there are significant deviations in the plane and height positions between the two. The comparison was carried out in different vegetation height zones (Figure 13). At the top of the mound, where the grass was mown and its height was only 1 to 2 cm, the maximum discrepancies between the UAV 3D model and the ground measurement isolines were 0.75 m for the plane position and 0.42 m for the height. On the slopes of the mound, where the height of the grass was between 5 and 10 cm, the maximum discrepancies between the plane position of the isolines were up to 0.41 m, and up to 0.42 m for the height. At the foot of the mound, where the height of the unheated grass was 60 to 100 cm, the maximum discrepancies between the plane position of the isolines reached 6.63 m, and up to 0.77 m for the height. The results of the discrepancies between the plane position of the isolines and their height are presented in Table 6.

The sharpness and contrast in aerial images are both becoming important issues for the use of UAV aerial imagery. Aerial image contrast problems occur in areas that fall under the shadow of trees or rough terrain on a sunny day. In this study, the image contrast of the aerial photographs was adjusted and, where necessary, increased. During the 2018 photofixation, the western and southwestern parts of the mound slope were in shadow. To highlight the terrain microforms in parts of the image on the southwest slope we used the brightness/contrast, shadows/highlights, color balance, hue/saturation, and photo files tools in Adobe Photoshop software.

The comparison of the large-scale maps of the Lepelionys mound surface created using the UAV 3D model or by using the ground topographic measurements, shows that the plane position of the isolines in the 3D model is highly micro-sinuous. This is due to the methods of isoline interpolation applied in the UAV 3D model, i.e., the calculation of the interfaces between multiple point pairs (about seven million pixel pairs) creates the non-continuous isolines.

Three-dimensional terrain modelling using UAV aerial imagery is currently expanding. The wider application of collaborative mapping initiatives in archaeology [54,55,56] will lead to an increasing use of nonprofessional UAV aerial imagery to identify undefined and unexplored archaeological sites from the 19th to the early 20th century.

## 5. Conclusions

The “Circle_3p” computer program designed by Artūras Bautrėnas, an associate professor of the Department of Cartography and Geoinformatics, employs the classical method of Delaunay and ensures a consistent systemic selection of points.The use of “Circle 3p” for the analysis of aerial photographs of the Lepelionys Mound has shown that the program needs to be improved by adding elements for the correction of the isolines.A comparison of the results of the geodetic measurements and the UAV images, showed that the best overlaps of surface microform isolines are on steep slopes. On the flat top of the mound surface, microform variance makes up 0.7 to 1.0 m. This is due to the rare density of the interpolation points calculated by the program “Circle_3p”. The variance of the isolines at the foot of the mound reaches 0.45 to 0.7 m (medium-height grass) and 0.95 to 1.35 m (high grass). The study has shown that external factors have a significant influence on the identification of the mound relief microforms.The research related to the GCPs position and shows that five GCPs arranged at the edges and the center of the object give the best accuracy as compared with other variations (three on the top, four on the edge, and ten GCPs).The unnatural curvature of isolines in the UAV 3D model, resulting in the abundance of unnatural surface microforms, is due to the interpolation techniques used to determine the isoline position, specifically the calculation of the interfaces between numerous point pairs.

## Figures and Tables

**Figure 1 sensors-19-05303-f001:**
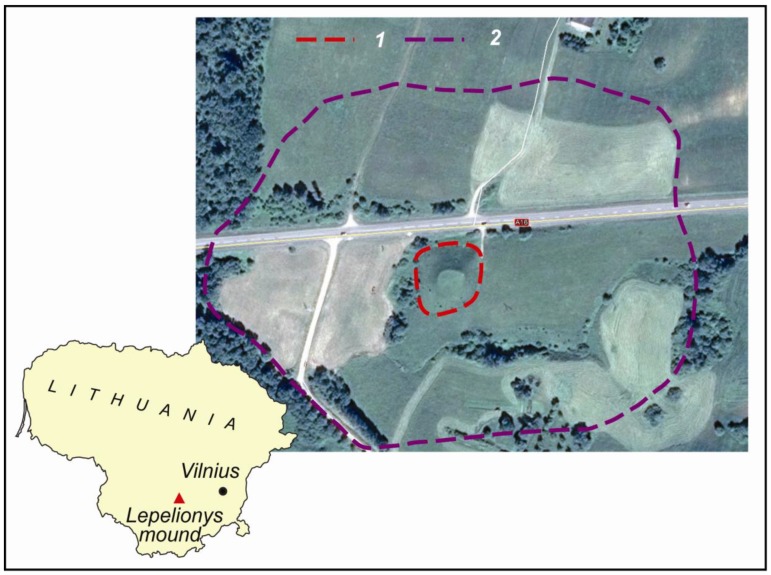
The location of Lepelionys Mound and the ancient settlement: (**1**) mound boundary and (**2**) ancient settlement territory boundary (according to V. Juškaitis [51]).

**Figure 2 sensors-19-05303-f002:**
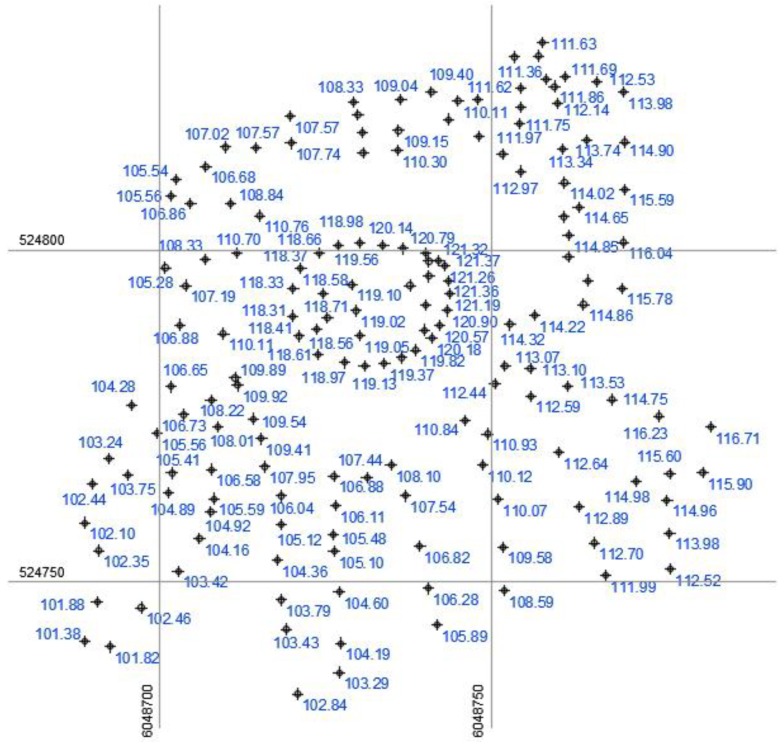
The selected points that were mapped to the LKS-94 coordinate system.

**Figure 3 sensors-19-05303-f003:**
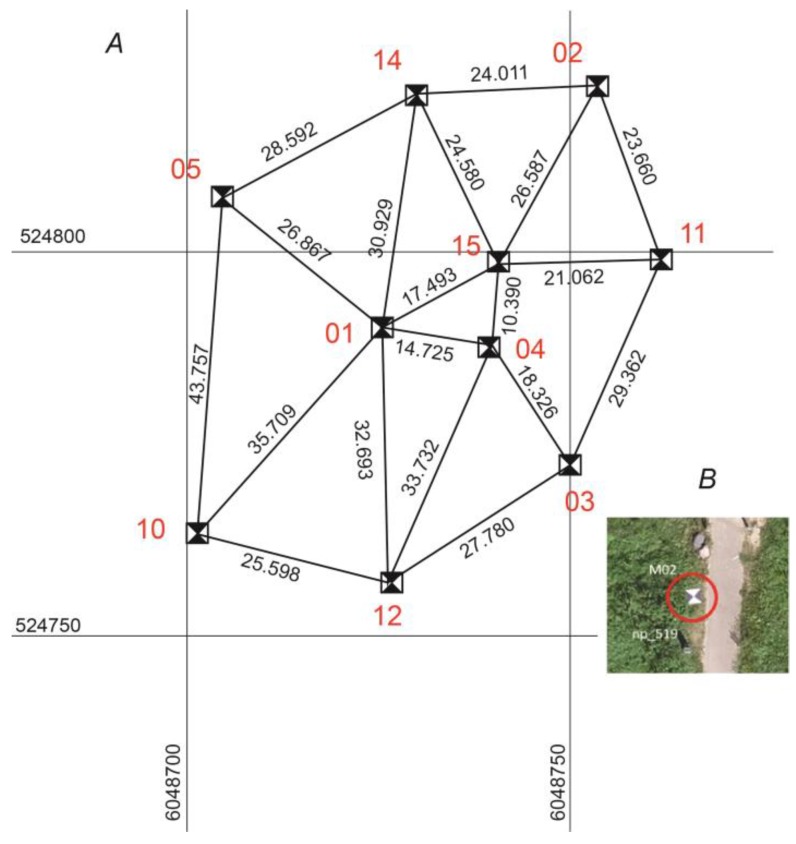
The ground control point marks (**A**) and the diagram of the ground control point (GCP) arrangement (**B**). The red circle defines the location of the ground mark.

**Figure 4 sensors-19-05303-f004:**
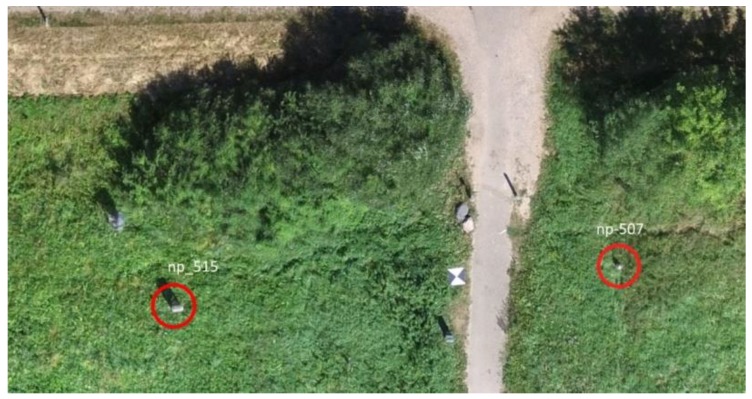
Examples of identified objects. The red circles define the small objects location, whose measured coordinates are used to adjust the 3D model.

**Figure 5 sensors-19-05303-f005:**
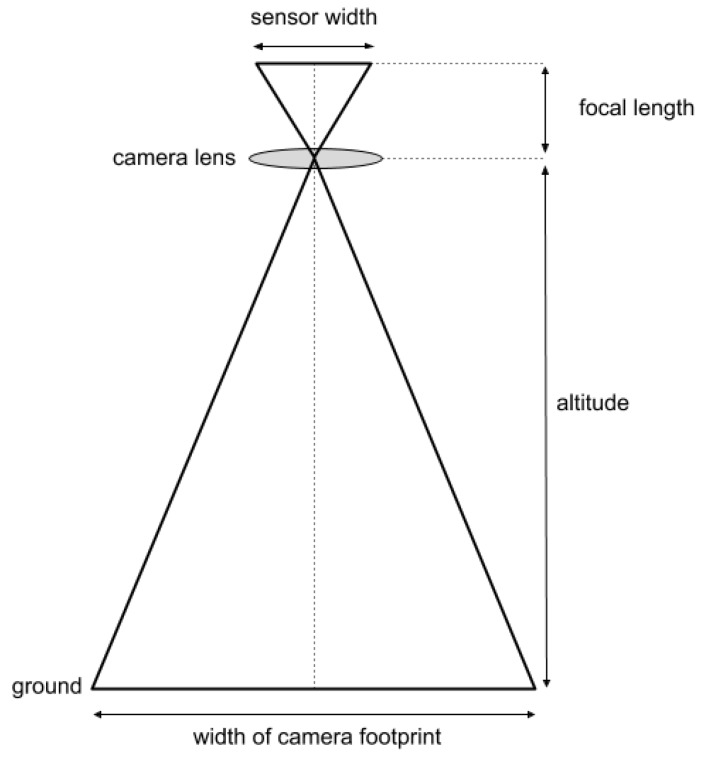
Visual representation of the nadir facing camera on the drone.

**Figure 6 sensors-19-05303-f006:**
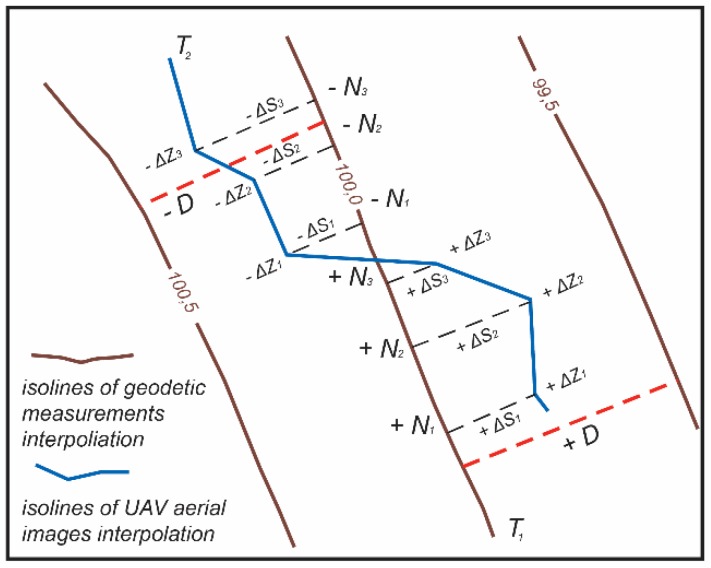
The scheme of elevation isoline position assessment using ground geodetic measurements and the mismatch between it and the aerial images from UAVs.

**Figure 7 sensors-19-05303-f007:**
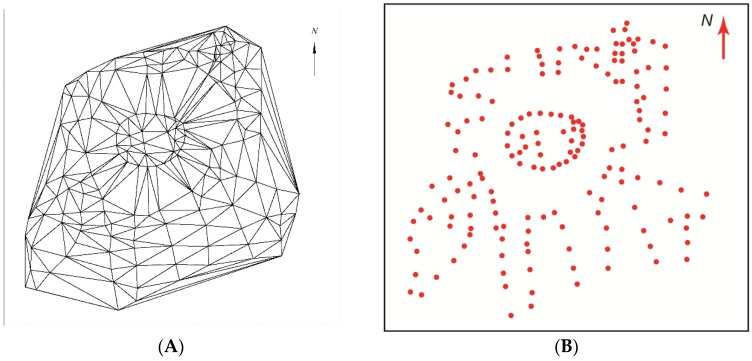
The Delaunay triangulation (**A**) among the 179 selected topographic points (**B**) using the program “Circle_3p”.

**Figure 8 sensors-19-05303-f008:**
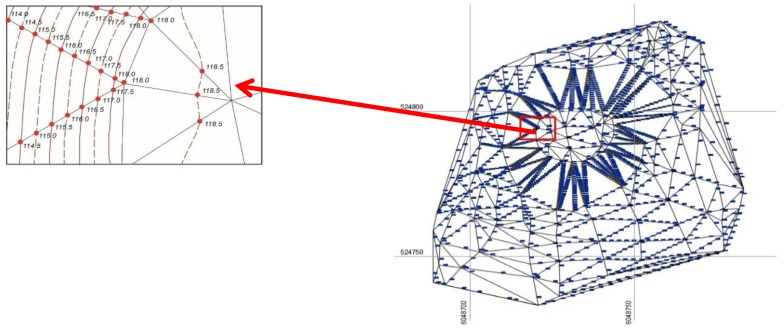
The interpolation among the vertices of selected triangles.

**Figure 9 sensors-19-05303-f009:**
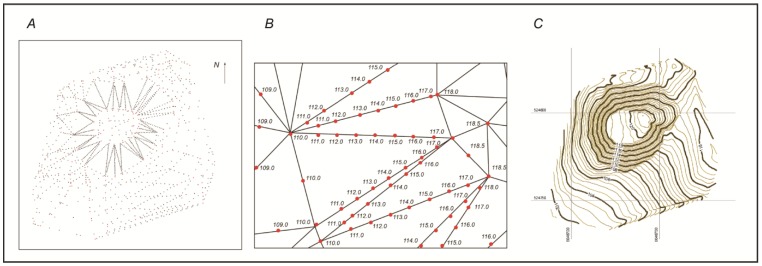
The points of interpolation (**A**), an interpolation sequence (**B**) and the relief isolines (**C**).

**Figure 10 sensors-19-05303-f010:**
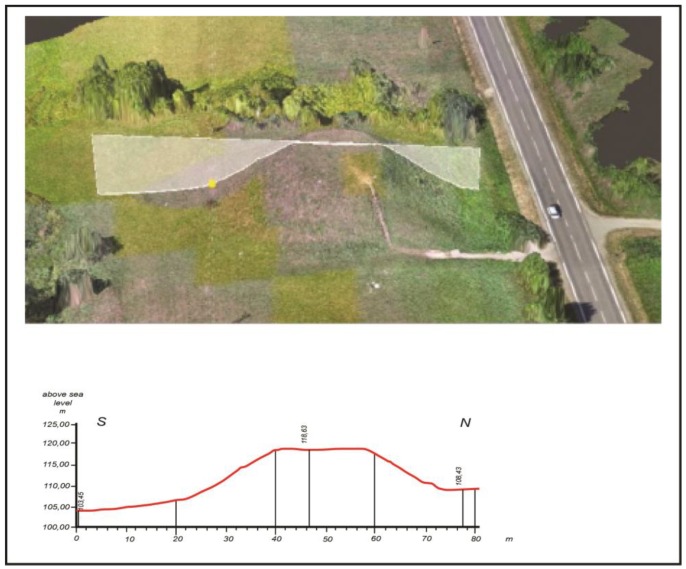
A cross-section of the Lepelionys Mound. The roughness in the red line indicates the remains of the former tree trunk fencing.

**Figure 11 sensors-19-05303-f011:**
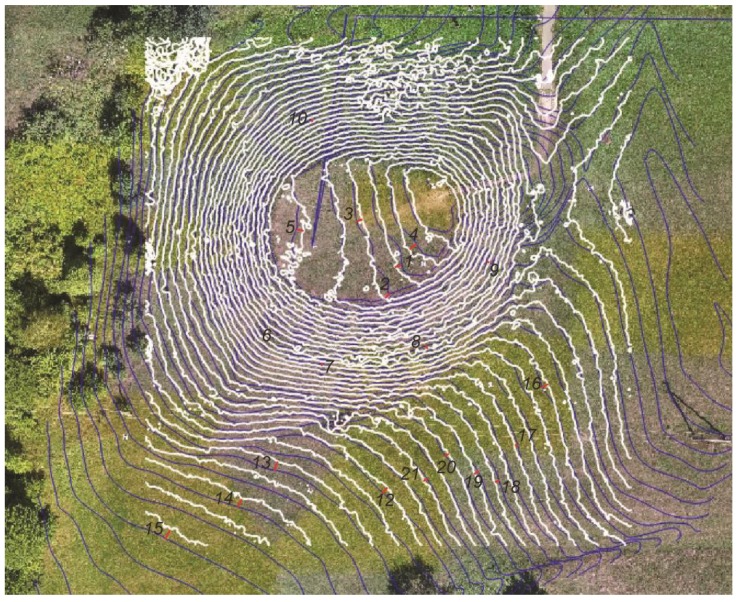
The variance of the equal height isolines in plane position.

**Figure 12 sensors-19-05303-f012:**
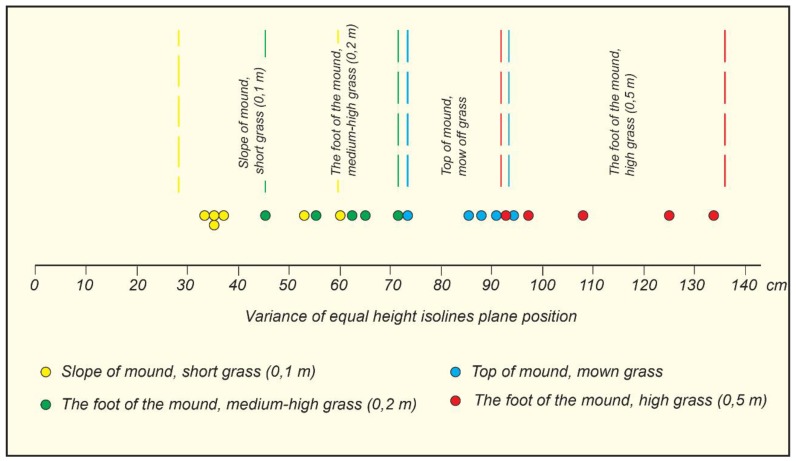
A diagram illustrating the parameters of variance of equal height isolines in plane position.

**Figure 13 sensors-19-05303-f013:**
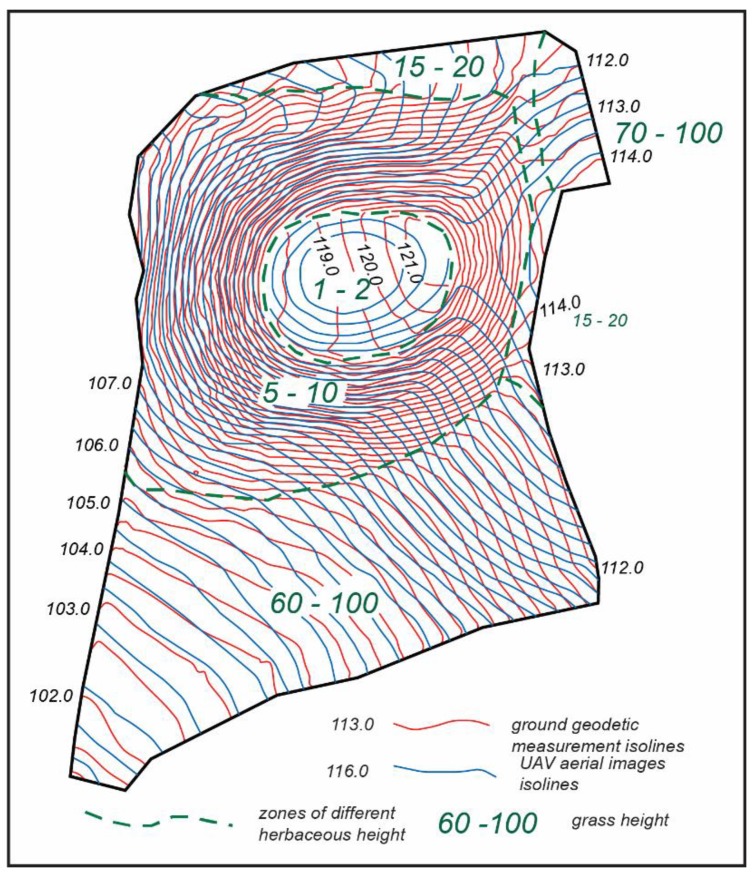
The zones of different herbaceous height on the Lepelionys mound.

**Table 1 sensors-19-05303-t001:** The specifications of unmanned aerial vehicle (UAV) DJI Inspire1 and the RGB Zenmus X3 camera.

Hovering Accuracy (GPS Mode)	Vertical: 0.5 m; Horizontal: 2.5 m
Max angular velocity	Pitch: 300°/s; Yaw: 150°/s
Max tilt angle	35°
Max ascent and descent speed	5 m/s; 4 m/s
Max speed	22 m/s
Max wind speed resistance	10 m/s
Max service ceiling above take-off point	120 m
Type and model	X3; FC350
Total and effective pixels	12.76 M; 12.4 M
Max capacity	64 GB
Maximal image size	4000 × 3000
ISO range	Photo- 100–1600; Video- 100–3200.
The electronic shutter speed	8 s–1/8000 s
A field of view (FOV)	94°
Supported file formats	Photo: JPEG, DNG; Video: MP4/MOV (MPEG-4 AVC/H.264)
Types of electronic media	Micro SD. Maximal capacity 64 GB. Class 10 or UHS-1
Sensor width (mm)	6.17
Focal length (mm)	4.55
Altitude (m)	50
Image width	4000
Image height	3000
GSD (cm/pixel)	1.695054945
Width (m)	67.8021978
Height (m)	50.85164835

**Table 2 sensors-19-05303-t002:** The parameters of variance of the equal height isolines in plane position.

Line No.	Variance of Equal Height Isolines in Plane Position, m	Slope Inclination, Degree in Brackets (Mean Slope Inclination Values)	Remarks
1	0.91	0	The top of the mound, mown grass
2	0.71	0	The top of the mound, mown grass
3	0.86	0	The top of the mound, mown grass
4	0.94	0	The top of the mound, mown grass
5	0.89	0	The top of the mound, mown grass
6	0.60	30–40	The slope of the mound, short grass (0.1 m)
7	0.36	30–40	The slope of the mound, short grass (0.1 m)
8	0.54	30–40	The slope of the mound, short grass (0.1 m)
9	0.36	30–40	The slope of the mound, short grass (0.1 m)
10	0.34	30–40	The slope of the mound, short grass (0.1 m)
11	0.36	30–40	The slope of the mound, short grass (0.1 m)
12	0.96	21–26	The foot of the mound, high grass (0.5 m)
13	1.26	21–26	The foot of the mound, high grass (0.5 m)
14	1.09	21–26	The foot of the mound, high grass (0.5 m)
15	1.32	21–26	The foot of the mound, high grass (0.5 m)
16	0.91	21–26	The foot of the mound, high grass (0.5 m)
17	0.45	14–19	The foot of the mound, medium-high grass (0.2 m)
18	0.67	14–19	The foot of the mound, medium-high grass (0.2 m)
19	0.71	14–19	The foot of the mound, medium-high grass (0.2 m)
20	0.54	14–19	The foot of the mound, medium-high grass (0.2 m)
21	0.63	14–19	The foot of the mound, medium-high grass (0.2 m)

**Table 3 sensors-19-05303-t003:** The errors of the measurements of the objects when using ten marks.

No.	GPS	DEM	Error Size (m)	Absolute Error Size (mm)
np-507				
X	6048772.162	6048772.216	−0.054	54
Y	524809.995	524810.072	−0.077	77
Z	112.138	112.212	−0.074	74
np-508				
X	6048771.963	6048771.971	−0.008	8
Y	524810.078	524810.066	0.012	12
Z	112.757	112.785	−0.028	28
np-515				
X	6048774.378	6048774.393	−0.015	15
Y	524804.426	524804.440	−0.014	14
Z	111.620	111.644	−0.024	24
np-522				
X	6048769.132	6048769.230	−0.098	98
Y	524803.655	524803.700	−0.045	45
Z	111.896	111.940	−0.044	44
np-530				
X	6048769.708	6048769.740	−0.032	32
Y	524793.519	524793.562	−0.043	43
Z	110.071	110.140	−0.069	69
np-560				
X	6048716.410	6048716.452	−0.042	42
Y	524751.812	524751.892	−0.080	80
Z	105.409	105.368	0.041	41
np-582				
X	6048712.964	6048712.978	−0.014	14
Y	524768.387	524768.400	−0.013	13
Z	106.038	106.006	0.032	32
np-586				
X	6048692.869	6048692.835	0.034	34
Y	524769.123	524769.100	0.023	23
Z	103.427	103.394	0.033	33
np-587				
X	6048692.729	6048692.734	−0.005	5
Y	524769.143	524769.154	−0.011	11
Z	104.131	104.107	0.024	24
np-596				
X	6048715.725	6048715.685	0.040	40
Y	524781.260	524781.290	−0.030	30
Z	107.443	107.399	0.044	44

**Table 4 sensors-19-05303-t004:** The absolute errors of objects when a 3D model is made on the basis of 10 marks with‚ ground control points.

Absolute Error (mm)			
Point No.	X	Y	Z
np-507	54	77	74
np-508	8	12	28
np-515	15	14	24
np-522	98	45	44
np-530	32	43	69
np-560	42	80	41
np-582	14	13	32
np-586	34	23	33
np-587	5	11	24
np-596	40	30	44

**Table 5 sensors-19-05303-t005:** A comparison of the absolute errors when different numbers of ground control points are used for a precise calculation. Values are in mm.

Point No.	3 Marks			4 Marks			5 Marks			7 Marks		
	x	y	z	x	y	z	x	y	z	x	y	z
np-507	402	608	514	309	194	263	56	80	51	48	70	73
np-508	86	105	56	36	42	42	11	12	26	10	9	19
np-515	175	193	64	78	14	10	15	8	25	13	16	26
np-522	95	145	574	48	33	254	95	56	57	88	39	49
np-530	954	657	1419	692	170	164	48	46	52	37	38	55
np-560	400	492	194	387	308	406	45	84	48	43	78	47
np-582	102	266	88	56	32	39	16	11	28	11	15	29
np-586	729	371	803	524	498	527	35	25	31	30	28	27
np-587	109	145	188	64	75	84	17	14	26	9	16	22
np-596	701	398	239	471	480	341	42	32	46	45	33	43

**Table 6 sensors-19-05303-t006:** The deviation between the ground measurement and the UAV isoline plane and height positions (data from June 2019).

Isoline Height	Number of Deviation ^a^		Maximum Plane Deviation, m ^b^		Average of Plane Deviation, m ^c^		Maximum of Height Deviation, m ^d^		Average of Height Deviation, m ^e^		Proportional Deviation ^f^
	−	+	−	+	−	+	−	+	−	+	
The foot of the mound. Height of the grass was 60–100 cm											
102.00	138	232	0.619	1.569	0.269	0.556	−0.109	+0.117	−0.051	+0.038	+0.005
102.50	741	399	6.653	2.527	1.539	0.834	−0.338	+0.214	−0.091	+0.071	−0.034
103.00	449	632	5.791	2.116	0.961	0.864	−0.778	+0.216	−0.110	+0.085	+0.004
103.50	377	431	3.950	1.663	0.569	0.698	−0.480	+0.186	−0.064	+0.079	+0.012
104.00	732	635	1.355	2.162	0.497	0.765	−0.167	+0.213	−0.064	+0.081	+0.003
104.50	882	863	2.301	1.822	0.787	0.569	−0.277	+0.293	−0.086	+0.080	−0.004
105.00	877	608	1.720	1.726	0.577	0.566	−0.190	+0.245	−0.063	+0.091	0.000
105.50	744	631	1.693	1.463	0.539	0.446	−0.199	+0.271	−0.074	+0.075	−0.005
106.00	536	356	1.629	0.902	0.449	0.276	−0.233	+0.139	−0.066	+0.048	−0.021
106.50	532	222	1.534	0.513	0.420	0.197	−0.282	+0.120	−0.077	+0.043	−0.042
107.00	318	251	0.771	0.959	0.310	0.293	−0.197	+0.179	−0.074	+0.059	−0.016
107.50	258	254	0.518	0.818	0.214	0.225	−0.126	+0.176	−0.047	+0.048	+0.000
108.00	203	303	0.681	0.799	0.265	0.253	−0.182	+0.183	−0.066	+0.060	+0.009
108.50	70	382	0.355	0.969	0.127	0.330	−0.109	+0.174	−0.037	+0.062	+0.047
109.00	416	242	1.021	0.508	0.401	0.135	−0.174	+0.147	−0.075	+0.040	−0.033
109.50	374	319	1.736	0.700	0.469	0.254	−0.406	+0.163	−0.115	+0.059	−0.035
110.00	212	305	0.776	0.415	0.250	0.166	−0.167	+0.090	−0.064	+0.036	−0.005
110.50	344	189	1.251	0.449	0.322	0.132	−0.238	+0.103	−0.066	+0.030	−0.032
111.00	449	336	1.401	0.575	0.412	0.211	−0.298	+0.199	−0.093	+0.064	−0.026
111.50	451	361	1.055	1.080	0.438	0.332	−0.251	+0.289	−0.103	+0.081	−0.022
112.00	537	206	1.843	0.779	0.585	0.270	−0.341	+0.141	−0.113	+0.048	−0.069
The slopes of the mound. Height of the grass was 5 to 10 cm											
110.00	138	137	0.422	0.695	0.205	0.325	0.197	+0.424	−0.096	+0.148	+0.027
110.50	112	158	0.349	0.477	0.159	0.195	0.197	+0.272	−0.086	+0.107	+0.027
111.00	162	124	0.326	0.3	0.114	0.109	0.184	+0.184	−0.064	+0.064	−0.006
111.50	169	118	0.355	0.358	0.154	0.133	−0.21	+0.234	−0.089	+0.088	−0.014
112.00	144	144	0.384	0.406	0.159	0.139	−0.247	+0.255	−0.01	+0.091	−0.001
112.50	146	137	0.347	0.381	0.138	0.137	−0.256	+0.246	−0.098	+0.096	−0.004
113.00	125	122	0.297	0.291	0.12	0.12	−0.227	+0.196	−0.09	+0.082	−0.002
113.50	115	113	0.272	0.265	0.124	0.116	−0.182	+0.189	−0.083	+0.081	+0.001
114.00	110	100	0.251	0.213	0.12	0.105	−0.161	+0.161	−0.08	+0.073	−0.007
114.50	103	108	0.277	0.288	0.123	0.132	−0.202	+0.207	−0.008	+0.01	+0.005
115.00	120	125	0.339	0.362	0.141	0.171	−0.199	+0.248	−0.092	+0.116	+0.013
115.50	115	139	0.378	0.329	0.181	0.137	−0.249	+0.219	−0.124	+0.099	−0.001
116.00	120	111	0.34	0.245	0.151	0.109	−0.257	+0.17	−0.108	+0.078	−0.018
116.50	95	124	0.206	0.237	0.095	0.097	−0.159	+0.155	0.072	+0.066	+0.007
117.00	119	88	0.267	0.222	0.119	0.108	−0.185	+0.16	−0.082	+0.078	−0.02
117.50	95	107	0.268	0.285	0.137	0.134	−0.168	+0.207	−0.085	+0.096	+0.009
118.00	91	100	0.271	0.274	0.132	0.131	−0.185	+0.176	−0.096	+0.082	−0.002
118.50	90	99	0.259	0.278	0.115	0.117	−0.193	+0.183	−0.009	+0.075	−0.002
119.00	110	68	0.27	0.242	0.122	0.112	−0.184	+0.158	−0.087	+0.075	−0.026
119.50	612	73	0.236	0.222	0.103	0.088	−0.159	+0.123	−0.079	+0.049	−0.009
120.00	87	92	0.254	0.405	0.094	0.123	−0.217	+0.177	−0.072	+0.063	−0.003
120.50	70	132	0.247	0.253	0.05	0.081	−0.245	+0.114	−0.041	+0.044	+0.014
121.00	57	50	0.116	0.159	0.063	0.069	−0.127	+0.127	−0.041	+0.071	+0.043
The top of the mound. Height of the grass was 1 to 2 cm											
121.00	71	151	0.468	0.443	0.260	0.240	−0.297	+0.055	−0.107	+0.031	−0.013
120.50	153	183	0.396	0.754	0.111	0.310	−0.070	+0.101	−0.021	+0.048	+0.017
120.00	86	251	0.373	0.466	0.148	0.269	−0.042	+0.091	−0.018	+0.051	+0.033
119.50	151	225	0.274	0.694	0.114	0.238	−0.048	+0.125	−0.018	+0.044	+0.019
119.00	205	395	0.736	0.585	0.256	1.228	−0.197	+0.419	−0.033	+0.114	+0.064
118.50	107	490	0.647	2.373	0.302	0.804	−0.235	+0.196	−0.106	+0.074	+0.042
121.00	71	151	0.468	0.443	0.260	0.240	−0.297	+0.055	−0.107	+0.031	−0.013
120.50	153	183	0.396	0.754	0.111	0.310	−0.070	+0.101	−0.021	+0.048	+0.017
120.00	86	251	0.373	0.466	0.148	0.269	−0.042	+0.091	−0.018	+0.051	+0.033
119.50	151	225	0.274	0.694	0.114	0.238	−0.048	+0.125	−0.018	+0.044	+0.019
119.00	205	395	0.736	0.585	0.256	1.228	−0.197	+0.419	−0.033	+0.114	+0.064
118.50	107	490	0.647	2.373	0.302	0.804	−0.235	+0.196	−0.106	+0.074	+0.042
121.00	71	151	0.468	0.443	0.260	0.240	−0.297	+0.055	−0.107	+0.031	−0.013
120.50	153	183	0.396	0.754	0.111	0.310	−0.070	+0.101	−0.021	+0.048	+0.017
120.00	86	251	0.373	0.466	0.148	0.269	−0.042	+0.091	−0.018	+0.051	+0.033
119.50	151	225	0.274	0.694	0.114	0.238	−0.048	+0.125	−0.018	+0.044	+0.019
119.00	205	395	0.736	0.585	0.256	1.228	−0.197	+0.419	−0.033	+0.114	+0.064
118.50	107	490	0.647	2.373	0.302	0.804	−0.235	+0.196	−0.106	+0.074	+0.042

* Explanation of the superscripts in the table: ^a^ number of negative (−N) and positive (+N) derivation, ^b^ maximum of negative (−Δ) and positive (+Δ) plane derivation, ^c^ average of negative (∑−Δ/−N) and positive (∑+Δ/+N) plane derivation, ^d^ maximum of negative (−Δz max) and positive (+Δz max) height derivation, ^e^ average of negative (∑−Δz/N_z_) and positive (∑+Δz/N_z_) height derivation, and ^f^ ratio of average positive/negative derivation and number of positive/negative derivation $\left(\frac{+\Delta }{-\Delta }\right)/\left(\frac{N+}{N-}\right)$.

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
