# Peer review of "A Comparison of the Influence of Vegetation Cover on the Precision of an UAV 3D Model and Ground Measurement Data for Archaeological Investigations: A Case Study of the Lepelionys Mound, Middle Lithuania"

_sensors, 2019, doi:10.3390/s19235303_

Round 1

Reviewer 1 Report

This paper offers a comparative study on the factors effecting the precision of the positions of estimated isolines. The research is based on a case study of an archaeological site and conducted with commercially available UAVs and photogrammetric software. The paper offers some practical guidelines of using these technical tools for making 3D on archaeological sites. 

There are several severe flaws in the presentation of this paper, however. 

Amass of papers are cited, but the literature survey is not informative at all. Readers have a difficult time of quickly understanding what have been done and what are the key research topics.  There are 5 statements in the conclusions. This reviewer had a hard time of identifying specific arguments or figures in the text to support these statements. For instance, for piont 3, what does it mean by "best overlaps" (line 314)? On line 310, is "consistent systematic selection" demonstrated by what?  On line 312, "adding the elements for correction" means? On line 320, "the edges and the centre of the object", what object? On line 324, what evidences support point 5? Part of the problems raised in 2 is due to the many sentences and captions that are incomprehensible. Line 125, 215, 174, 216 (horizontal interpolation), 244, 260, 266, 267, 270, and 272. Equations are printed in poor quality. Fig. 12 (line 230) seems to be mis-referred. So are Fig. 2 (line 235) and Fig. 3 (line 237).    The paper seems to identify two sources of errors in estimating isolines' positions. The first is the heights in the 3D model obtained by aerial photos, mainly effected by the grass. The second is the interpolation by the program Circle_3p. But the abstract and conclusions sections deliver the message in a vague way and the readers don't know the relative degrees of contribution of the two factors.  It seems to me that if access to the algorithms inside the "Pixoprocessing" software is available, many problems can be discussed in a more efficient and effective manner. Without it, many conclusions in this paper are quite derivative, and if the software undergoes version updates, maybe many statements in this paper will no longer be valid. This can seriously damage the value of this paper. 

Author Response

Dear Reviewer.

Thank You for the comments.

We took You comments into account in correcting the errors.

We also comment again on what we sought in this study.

Our comments

When measuring archaeological objects with the UAV aerial photographs, which are covered with grass vegetation (in grass vegetation period), a certain error of surface fixation error is possible. In our case, for example, these errors are 0.5 m and more. The error can be determined by measuring grass at several locations in the archaeological site and deriving an average height. In this way, it is possible to recognize micro-elevation or micro-depressions and other terrain features.

The task of the study is to determine the possibility of using UAVs as a tool for recognition of large area archaeological sites. Often, existing archaeological sites have much larger boundaries, but complicated relief, vegetation cover and other surface features make it difficult to identify it’s complexly. Ground topographic measurements are expensive and time consuming. As a result, relief microforms are possible by introducing corrections that are focused on vegetation / grass height.

Based on the investigation results, we mind that the inaccuracy of the UAV is relatively high when investigating archaeological objects, but aerial photographs capture all surface irregularities and provide an overall view. Such an image can be used to more clearly identify specific local study sites and create topographic plans for them. Such a methodology saves money and time.

Comparing the UAV result map with a ground measurement large scale topographic map (plan) shows that the accuracy of the topographic plan is one level higher. However, topographic plans often do not form an integral picture of the whole, which is very important for the exploration of an unknown archaeological site.

Responses to concrete comments

1 For instance, for point 3, what does it mean by "best overlaps" (line 314)?

It had in mind that interpolated isolines was drawn on the basis of terrestrial topographic measurements and from UAVs best coincide on steep slopes.

2. On line 312, "adding the elements for correction" means?

The comparisons of the isolines position have shown that the “Circle_3p” software used requires further refinement to better identify isolines mismatches.

3.On line 320, “edges and centre of object,” what object?

The object in question is a mound with coordinated marks at the it top and at the foot.

4. On line 324, what evidences support point 5?

The claim in conclusion 5 is supported by the data in Tables 2 and 6.

5. Equations are printed in poor quality.

The equations are reprinted using Equation tool of Word programme.

6. Fig. 12 (line 230) seems to be mis-referred.

Thank you for your note. Really must be Fig. 11. The mistake is corrected.

7. The paper seems to identify two sources of errors in estimating isolines' positions. The first is the heights in the 3D model obtained by aerial photos, mainly effected by the grass. The second is interpolation by the program Circle_3p. But the abstract and conclusion sections deliver the message in a vague way and the readers don't know the relative degrees of contribution of the two factors. It seems to me that if you access the algorithms inside the “Pixoprocessing” software is available, many problems can be discussed in a more efficient and effective manner. Without it, many conclusions in this paper are quite derivative, and if the software undergoes version updates, maybe many statements in this paper will no longer be valid. This can seriously damage the value of this paper.

The aim of this study was to compare the position of isolates from topographic measurements and UAV aerial photographs. Let us assume that topographic measurements give more accurate results. Circle 3 p software for interpolation of isolines based on Delaunay triangulation method. This method is characterized by greater accuracy.

We are sorry, but we don't have access to “Pixprocessing” algorithms.

Also, we inform You that the article has been submitted English-language for proofread of native English linguist.

The article will be reviewed next week. We will send You the revised form as soon as possible.

With respect.

Algimantas Česnulevičius

Reviewer 2 Report

This paper descibes the case of measurement of specific site, which allow to compare measurement accuracy with different height of grass. Based on this experiment the authors claim that sufficient accuracy cannot be reached if grass is 0.5 meters height. The main question is novelty and originality of the work. The precision of photogrammetric-based measurement from UAV has been measured before on many sites. One of the main thing is that photogrammetric software allows creation of DSM, not DEM. If difference in height of the grass is taken into account that the main result becomes rather obvious. In current state the paper is more a technical report, not a paper regarding some original contribution, be it some new facts or methods. The originality of the work should be specifically addressed and compared to that of similar methods, which measures the accuracy of UAVs. Also specific target precision should be specified, so it would be better understandable to reader what kind of precision is enough for finding such kind of historic sites.

Author Response

Dear Reviewer.

Thank You for the comments.

We took You comments into account in correcting the errors.

We also comment again on what we sought in this study.

Our comments

When measuring archaeological objects with the UAV aerial photographs, which are covered with grass vegetation (in grass vegetation period), a certain error of surface fixation error is possible. In our case, for example, these errors are 0.5 m and more. The error can be determined by measuring grass at several locations in the archaeological site and deriving an average height. In this way, it is possible to recognize micro-elevation or micro-depressions and other terrain features.

The task of the study is to determine the possibility of using UAVs as a tool for recognition of large area archaeological sites. Often, existing archaeological sites have much larger boundaries, but complicated relief, vegetation cover and other surface features make it difficult to identify it’s complexly. Ground topographic measurements are expensive and time consuming. As a result, relief microforms are possible by introducing corrections that are focused on vegetation / grass height.

Based on the investigation results, we mind that the inaccuracy of the UAV is relatively high when investigating archaeological objects, but aerial photographs capture all surface irregularities and provide an overall view. Such an image can be used to more clearly identify specific local study sites and create topographic plans for them. Such a methodology saves money and time.

Comparing the UAV result map with a ground measurement large scale topographic map (plan) shows that the accuracy of the topographic plan is one level higher. However, topographic plans often do not form an integral picture of the whole, which is very important for the exploration of an unknown archaeological site.

Responses to concrete comments

1. One of the main thing is that photogrammetric software allows creation of DSM, not DEM.

Thank You for the note. We have changed the DEM to DSM.

Also, we inform You that the article has been submitted English-language for proofread of native English linguist.

The article will be reviewed next week. We will send You the revised form as soon as possible.

With respect.

Algimantas Česnulevičius

Reviewer 3 Report

The aim of this research was to conduct a comparative analysis of precision in ground  geodetic versus the 3D measurements from UAV while establishing the impact of herbaceous vegetation on the UAV 3D model.Ground measurements were obtained using Trimble GPS and UAV “Inspire 1” was used for taking aerial photographs. Following the data of ground measurements and aerial photographs, large scale surface maps were drawn and the errors in the measurement of the position of isolines were compared. The results showed that the largest errors in the positional measurements of fixed objects were conditioned by the height of grass.

Author Response

Dear Reviewer.

Thank You for the comments.

We took You comments into account in correcting the errors.

We also comment again on what we sought in this study.

Our comments

When measuring archaeological objects with the UAV aerial photographs, which are covered with grass vegetation (in grass vegetation period), a certain error of surface fixation error is possible. In our case, for example, these errors are 0.5 m and more. The error can be determined by measuring grass at several locations in the archaeological site and deriving an average height. In this way, it is possible to recognize micro-elevation or micro-depressions and other terrain features.

The task of the study is to determine the possibility of using UAVs as a tool for recognition of large area archaeological sites. Often, existing archaeological sites have much larger boundaries, but complicated relief, vegetation cover and other surface features make it difficult to identify it’s complexly. Ground topographic measurements are expensive and time consuming. As a result, relief microforms are possible by introducing corrections that are focused on vegetation / grass height.

Based on the investigation results, we mind that the inaccuracy of the UAV is relatively high when investigating archaeological objects, but aerial photographs capture all surface irregularities and provide an overall view. Such an image can be used to more clearly identify specific local study sites and create topographic plans for them. Such a methodology saves money and time.

Comparing the UAV result map with a ground measurement large scale topographic map (plan) shows that the accuracy of the topographic plan is one level higher. However, topographic plans often do not form an integral picture of the whole, which is very important for the exploration of an unknown archaeological site.

Also, we inform You that the article has been submitted English-language for proofread of native English linguist.

The article will be reviewed next week. We will send You the revised form as soon as possible.

With respect.

Algimantas Česnulevičius

Round 2

Reviewer 1 Report

The authors do answer most of my questions, but they didn't modify the article accordingly. My questions also meant to make the article easier for readers to understand. So the answers should be properly fit into the text. Also, I think the authors should make clear in the article the limitation of their approach, especially since they don't have access to the inner mechanism of Pix4D, their conclusions are highly derivative, i.e., the conclusions may no longer hold if the software gets updated. Otherwise, I think the article is acceptable for publishing. 

Author Response

Dear Reviewers and Editorial.

I am sending You a adjusted and revised version of our article after proof-reading. The review was conducted by Proof-Reading-Service: Devonshire Business Center, Works Road, Letchworth Garden City, SG6 1GJ, Hertfordshire, United Kingdom.

Proof-reading identification subject: sensors-625919. after correction and addition (ref. no. 201911-1804842).

We believe that after Your comments and proof-reading review, the article has become clearer to journal readers.

Thank You again for Your comments

With respect

Algimantas Česnulevičius
